# Evaluation of Visual and Patient—Reported Outcomes, Spectacle Dependence after Bilateral Implantation with a Non-Diffractive Extended Depth of Focus Intraocular Lens Compared to Other Intraocular Lenses

**DOI:** 10.3390/jcm11175246

**Published:** 2022-09-05

**Authors:** Anna Dołowiec-Kwapisz, Halina Piotrowska, Marta Misiuk-Hojło

**Affiliations:** 1Department of Ophthalmology, Hospital in Zgorzelec, 59-900 Zgorzelec, Poland; 2Department of Ophthalmology, Wrocław Medical University, 50-556 Wrocław, Poland

**Keywords:** cataract surgery, EDOF, intraocular lens, presbyopia, multifocal lens, photic phenomena

## Abstract

**Purpose**: To evaluate postoperative outcomes, spectacle dependance and the occurrence of the photic phenomena in patients after cataract surgery following the implantation of a non-diffractive extended depth of focus (EDOF) intraocular lens was compared to monofocal and multifocal lenses. **Methods****:** We enrolled patients with bilateral cataracts who wanted to reduce their dependence on glasses in the study. They were followed for 6 months. The study group in which the EDOF lens was implanted consisted of 70 eyes in 35 patients. The control groups consisted of: 52 eyes in 26 patients in whom a multifocal was implanted and 52 eyes in 26 patients with implanted monofocal lens. After a total of 2 weeks, 2 months and 6 months post-surgery the following were evaluated: uncorrected and corrected visual acuity at 4 m, 80 cm, 40 cm, manifest refraction expressed as mean refractive spherical equivalent (MRSE), contrast sensitivity, intraocular pressure. A questionnaire on independence from ocular correction, the occurrence of photic phenomena, and patient satisfaction was also completed. **Results**: Monocular and binocular visual acuity and MRSE 6 months after the procedure were compared between three groups. All of the main analyses, except for comparisons of uncorrected distance visual acuity (both monocular and binocular) level, were significant. Contrast sensitivity was lower among patients with multifocal lens than among patients with EDOF lens. Halo and glare after 6 months were seen more often among patients with multifocal lens than among patients with the other lens (65% of eyes with multifocal lens vs. 6% of eyes with EDOF lens and 0% of eyes with monofocal lens). Glasses were needed by 35% of patients with EDOF lens, and by 96% of patients with monofocal lens and in none of the patients with multifocal lens. **Conclusions**: Most patients qualify for the implantation of a non-diffractive EDOF lens. Post-operative visual acuity improves at any distance. The best monocular visual acuity for intermediate distances is provided by an EDOF lens, and for near distance by a multifocal lens. The EDOF lens definitely increases independence from spectacle correction compared to monofocal lenses; however, the greatest degree of independence from spectacles is provided by multifocal lenses. The incidence of photic phenomena is slightly higher than that of a monofocal lens, and much lower for a multifocal lens.

## 1. Introduction

Cataract surgery is one of the most frequently performed procedures around the world. Intraocular lenses (IOLs) have developed significantly in recent years. IOLs are used in cataract surgery to replace a cloudy lens and in refractive lens exchange (RLE). The most common implanted lenses are monofocal lenses, which provide good acuity to one type of vision correction, mainly for a far distance. In addition to monofocal lenses, premium lenses, which have a more advanced structure and different optical properties are available. These lenses correct presbyopia, i.e., insufficient accommodation that occurs physiologically after the age of 40. Premium lenses include multifocal intraocular lenses (MIOLs), extended depth of focus lenses (EDOF) and accommodative lenses. These lenses improve visual acuity after cataract surgery and allow for full or partial independence from spectacle correction. Both the extension of life expectancy, lifestyle changes and greater professional activity of the elderly contribute to the willingness to become independent from eyeglass correction not only in the distance, but also in the near and intermediate distance [1].

MIOLs allow for the greatest degree of independence from eyeglass correction, but they have a lower contrast sensitivity and a higher rate of photic phenomena, such as halo and glare. The eligibility criteria for implantation in this group of lenses are the most stringent, and the eyes should be free of any pathology so that patients can achieve the best possible postoperative results [2].

EDOF lenses can be positioned between monofocal and multifocal lenses. They provide good uncorrected distance and intermediate visual acuity; however, visual acuity without near correction may be insufficient. They work by creating a single, elongated focus to increase the depth of field. The elongated focus is designed to eliminate the close-up and distance overlap that occurs with multifocal lenses, thus eliminating the halo effect. In addition, EDOF lenses provide a continuous focus range, without the power distribution being unevenly divided, and thus avoiding secondary out-of-focus images [3,4]. Compared to multifocal lenses, they do not lower contrast sensitivity and cause less dysphotopsia [5].

There are different methods of creating these lenses. One of them is the use of spherical aberration, however, it differs from patient to patient in the population and is influenced by pupil width [6,7]. Another way is to use diffraction optics to obtain the EDOF effect. However, this can lead to the development of a dysfunction similar to those seen after multifocal lens implantation, and uncorrected near vision acuity may not be satisfactory [8]. Another method may be the use of a circular mask, as with the IC-8 lens, a small-aperture EDOF lens. However, this can reduce the amount of light entering through the diaphragm and is typically used in the non-dominant eye [9].

An increasing number of patients presenting for cataract surgery want to be independent from eyeglass correction, and at the same time are afraid of the photic phenomena after the procedure. There has been a growing number of patients after refractive surgery in the past who would like to regain independence from eyeglass correction and do not qualify for multifocal lens implantation. The same applies to patients with ocular diseases who would like to choose premium lenses. For these patients, EDOF lenses provide a chance to improve uncorrected acuity at all distances. Due to the fact that the EDOF Vivity lens (Alcon Laboratories, Inc., Fort Worth, TX, USA) has a unique, non-diffractive optical part; it allows patients to see well from near to distance. It is based on the non-diffractive X-wave technology, which modifies the wave front and creates one elongated focus without splitting the light. Thanks to these properties, the lens reduces the risk of dysphotopsia and does not worsen the contrast sensitivity. In the construction of the lens, two zones can be distinguished: the transition zone 1 is responsible for stretching the wave front and creating a continuous elongated focus, while the transition zone 2 is responsible for shifting the wave front from hyperopia to short-sighted in order to use all the light energy. It has 1.5 D defocusing and negative asphericity of the anterior surface (−0.2 μm) [10,11,12]. Moreover, the extended depth of focus, such as that seen in the Vivity lens, can “forgive” the imperfection of IOL power selection caused by the difficulty in calculating IOL power (especially in patients who have undergone refractive surgery in the past) [10].

To date, few articles have been published on postoperative outcomes in patients with non-diffractive EDOF lens implantation, including one in patients with ocular pathologies. This paper additionally includes patients with a history of refractive surgery, who will be increasingly more numerous in the future and would like to choose a premium lens. To the best of our knowledge, there are currently no publications on the comparison of the Vivity lens with monofocal and multifocal lenses. This work may provide some insight into lens selection, especially for patients with ocular pathologies and after refractive surgery who are ineligible for MIOLs or are worried about their side effects. The aim of the article is to evaluate the postoperative results, spectacle dependence, the occurrence of the photic phenomena in patients after cataract surgery using a non-diffractive EDOF–Vivity intraocular lens compared to multifocal and monofocal intraocular lenses.

## 2. Patients and Methods

### 2.1. Study Design

This single-center, prospective, comparative study was conducted at the Ophthalmology Department of the Hospital in Zgorzelec in line with the Helsinki Declaration and approved by the Bioethics Committee at the Medical University of Wroclaw. Written consent was obtained from all patients.

### 2.2. Study Population

The study group in which the Vivity lens was implanted (DFT015 or the toric version of the lens-called the EDOF group) consisted of 70 eyes in 35 patients. The control groups consisted of 52 eyes in 26 patients in whom a Panoptix multifocal lens (TNTFOO or the toric version of the lens-called the MULTI group) was implanted and 52 eyes in 26 patients with implanted monofocal lens (SA60WF or the toric version SN6AT3-7-called the MONO group). All lenses implanted in the patients involved in our study are single-piece, aspheric, are constructed of the same material-hydrophobic, and are based on the same platform-Acrysof (Alcon Laboratories, Inc., Fort Worth, TX, USA) [10,11,13,14,15]. The EDOF and multifocal lens were donated by the Alcon Company for the purpose of this study.

The study included patients aged 35–75 diagnosed with bilateral cataracts in whom the removal of the cataract was planned by phacoemulsification.

Exclusion criteria included: patients under the age of 35, over the age of 75, pregnant, after a corneal transplant, with a history of past eye injuries, diseases of the anterior and posterior segment of the eye that may have significantly reduced the quality of vision after surgery, such as: advanced glaucomatous neuropathy, advanced diabetic retinopathy, amblyopia, corneal scarring and dystrophy, exudative age-related macular degeneration (AMD), post-posterior vitrectomy condition or elective surgery, and clinically significant severe dry eye syndrome. Patients after cerebral events that could have affected visual acuity were also excluded from the study.

### 2.3. Preoperative Assessment

The pre-operative examination consisted of: anterior and posterior segment examination in a slit lamp, intraocular pressure examination, refraction examination (NIDEK ARK-510A-Nidek Co. Ltd., Gammagori, Aichi, Japan), monocular visual acuity examination in logarithm of the minimum angle of resolution (logMAR) scale: uncorrected distance visual acuity (UCDVA) at 4 m, best corrected distance visual acuity (BCDVA), uncorrected intermediate visual acuity (UCIVA) at 80 cm, best corrected intermediate visual acuity (BCIVA), uncorrected near visual acuity (UCNVA) with 40 cm, best corrected near visual acuity (BCNVA), monocular contrast sensitivity at 40 cm (Pelli-Robson test, GIMA charts, Gessate, Italy), biometry using an Argos SS-OCT optical biometer (Movu, Inc., Kamaki, Japan) and IOL Master 500 (Carl Zeiss Meditec AG, Jena, Germany), Oculazer ™ WaveLight^®^ II corneal tomography and topography (Alcon Laboratories, Inc., Fort Worth, TX, USA), posterior segment optical coherence tomography (OCT) (OCT III, Carl Zeiss Meditec AG, Jena, Germany). A standardized ETDRS chart at 4 m, 80 cm, and 40 cm was used to measure visual acuity (VA).

In the study, the refractive result was written as a spherical equivalent, defined as the sum of the spherical power and half of the cylindrical power [16]. The results obtained were classified as myopia, emmetropia or hyperopia. For myopia there was a spherical equivalent less than −0.5 D, for emmetropia-a spherical equivalent in the range of −0.5 and +0.5 D, and in the case of hyperopia, a spherical equivalent greater than +0.5 D. This division was adopted in accordance with other large cross-sectional and dynamic studies [16,17,18].

### 2.4. Postoperative Assessment

Controls were performed 2 weeks, 2 months (6–8 weeks) and 6 months after cataract surgery. Controls included: anterior and posterior segment examination in a slit lamp, an intraocular pressure test, manifest refraction expressed as mean refractive spherical equivalent (MRSE), monocular visual acuity test in the logMAR scale: UCDVA, BCDVA, UCIVA, BCIVA, UCNVA, BCNVA, monocular contrast sensitivity and the postoperative questionnaire.

Six months after the second eye surgery, binocular visual acuity was assessed: UCDVA, UCIVA, UCNVA.

Measurements were performed under photopic conditions (250–300 lumens/mm^2^) in all cases. The preoperative and postoperative examinations were carried out by the same person.

### 2.5. Subjective Visual Quality Questionnaire

Patients were asked “yes/no’’ questions regarding independence from glasses, the occurrence of postoperative dysphotopsia such as halo, glare, starburst, was assesed on a scale from 1 to 5 in increments of 1 (1-slight, 5-very high). In addition, patients were asked to rate their satisfaction with the procedure (scale from 1 to 5).

### 2.6. Surgical Technique

All cataract surgery with implantation of an appropriate intraocular lens was performed by the same surgeon (H.P.). All the procedures were uneventful. They were performed under drip (Alcaine) and intraocular (Mydriane) anesthesia. The lenses were implanted through a 2.2 mm corneal incision into the lens bag. At the end of the procedure, in accordance with European standards, cefuroxime solution was administered into the anterior chamber.

Implant power for the Vivity lens was calculated on an Argos optical biometer using Barret’s formula, setting postoperative results to either emmetropia (18 patients, 36 eyes) or minimonovision (17 patients, non-dominant eye set to target ~ −0.75D). Eye dominance was determined by taking the Mile’s test. For Panoptix lenses, the Argos biometer and Barret’s formula were used, setting postoperative results to emmetropia. Implant power of monofocal lenses was calculated using an IOL Master 500 or Argos biometer, using the SRK/T formula or the Barret formula by setting the target to emmetropia. With axial length <22 mm, the Haigis formula was used. In the case of patients after previous radial keratotomy, measurements were made on an Argos biometer using the Barret true K formula.

### 2.7. Statistical Analysis

All analyses were performed in statistical environment R, version 4.1.3. Quantitative variables were compared between groups with the Kruskal-Wallis test or with ANOVA analysis and between measurements—with the Friedman’s test (for more than two measurements) or Wilcoxon’s test for dependent samples (for two measurements). These tests were chosen because all variables’ distributions significantly differed from the normal distribution (checked with Shapiro-Wilk’s test). Median differences with 95% confidence intervals were given when comparing two measurements. Dependencies for qualitative variables were analyzed with the chi-square test or the Fisher’s exact test. Significance level in the analysis equalled α = 0.05.

## 3. Results

Pre- and postoperative data of 174 eyes (87 patients) were included in the analysis. A total of 35 patients (70 eyes) received the Vivity IOL (19 toric and 51 non-toric), 26 patients (52 eyes) underwent the implantation of monofocal (12 toric and 40 non-toric) lens, and 26 patients (52 eyes) the PanOptix lens (10 toric and 42 non-toric).

### 3.1. Demographics and Preoperative Data

Detailed demographic characteristics, biometry values, mean preoperative refractive errors and visual acuity (VA), contrast sensitivity, and intraocular pressure from the three groups are presented in Table 1.

Among the MULTI group there was a smaller proportion of eyes with myopia than in the other groups (23% vs. 44% for EDOF and 52% for MONO) and a higher proportion of eyes with emmetropia (17% vs. 7% for EDOF and 4% for MONO) or with hyperopia (60% vs. 49% for EDOF and 44% for MONO), *p* = 0.012. The analysis comparing quantitative variables was significant for: UCDVA (*p* = 0.017), BCDVA (*p* = 0.004), BCNVA (*p* = 0.014) and ACD (*p* = 0.012). Post-hoc analyses showed that patients from the MULTI group had a lower level of UCDVA variable than EDOF and MONO groups and lower level of BCNVA than the MONO group. Patients from the MONO group were characterized by a higher level of BCDVA than two remaining groups and by a lower level of ACD than patients from EDOF group (*p* < 0.050 for all post-hoc analyses), Table 1.

### 3.2. Refractive and Visual Outcomes

Monocular and binocular visual acuity and MRSE 6 months after the procedure were compared between EDOF, MONO and MULTI groups. All main analyses, except for comparisons of UCDVA (both monocular and binocular) level, were significant (*p* < 0.050). Post-hoc analyses showed that the level of: MRSE, BCDVA and BCIVA was higher among MULTI group than among EDOF group (*p* < 0.010 for all post-hoc analyses). Contrast sensitivity was lower among MULTI group than among EDOF group. The level of UCIVA (both monocular and binocular visual acuity) was higher in the MONO group than in the EDOF and MULTI groups (*p* < 0.001 for all post-hoc analyses) and higher in the MULTI group than in the EDOF group (*p* < 0.050 for both post-hoc analyses). The level of UCNVA (both monocular and binocular visual acuity) was higher in the MONO group than in the EDOF and MULTI groups (*p* < 0.001 for all post-hoc analyses) and higher in the EDOF group than in the MULTI group (*p* < 0.001 for both post-hoc analyses). The level of BCNVA was higher among MULTI group than among MONO group (*p* = 0.007), Table 2.

When the EDOF group is divided into two subgroups: (1) patients with target set to emmetropia, (2) patients with minimonovision (non-dominant eye set to target ca. −0.75 D) the results of binocular uncorrected visual acuity 6 months after surgery are as follows, Table 3.

#### Refractive and Visual Outcomes in Baseline and after 6 Months in Each Group

In the EDOF group at the baseline the level of: UCDVA (MD 95% CI = 0.70 (0.63; 0.82); *p* < 0.001), BCDVA (MD 95% CI = 0.25 (0.30; 0.35); *p* < 0.001), UCIVA (MD 95% CI = 0.75 (0.64; 0.80); *p* < 0.001), BCIVA (MD 95% CI = 0.20 (0.25; 0.35); *p* < 0.001), UCNVA (MD 95% CI = 0.40 (0.45; 0.60); *p* < 0.001), BCNVA (MD 95% CI = 0.30 (0.25; 0.40); *p* < 0.001) and IOP (MD 95% CI = 2.85 (1.90; 3.00); *p* < 0.001) was higher than after 6 months. The level of contrast was lower at the baseline than after 6 months (MD 95% CI = −0.20 (−0.35; −0.20); *p* < 0.001).

In the MONO group almost all comparisons between baseline and after 6 months were significant (except of the MRSE level). At the baseline the level of: UCDVA (MD 95% CI = 0.70 (0.55; 0.75); *p* < 0.001), BCDVA (MD 95% CI = 0.25 (0.30; 0.38); *p* < 0.001), UCIVA (MD 95% CI = 0.30 (0.30; 0.50); *p* < 0.001), BCIVA (MD 95% CI = 0.30 (0.24; 0.33); *p* < 0.001), UCNVA (MD 95% CI = 0.20 (0.13; 0.33); *p* < 0.001), BCNVA (MD 95% CI = 0.40 (0.30; 0.40); *p* < 0.001) and IOP (MD 95% CI = 2.80 (0.85; 2.45); *p* < 0.001) was higher than after 6 months. The level of contrast sensivity was lower at the baseline than after 6 months (MD 95% CI = −0.30 (−0.35; −0.25); *p* < 0.001).

All analyses were significant in the case of the MULTI group. At the baseline the level of: MRSE (MD 95% CI = 0.87 (0.00; 1.37); *p* = 0.043), UCDVA (MD 95% CI = 0.40 (0.40; 0.57); *p* < 0.001), BCDVA (MD 95% CI = 0.25 (0.25; 0.28); *p* < 0.001), UCIVA (MD 95% CI = 0.60 (0.60; 0.80); *p* < 0.001), BCIVA (MD 95% CI = 0.20 (0.15; 0.25); *p* < 0.001), UCNVA (MD 95% CI = 0.70 (0.65; 0.80); *p* < 0.001), BCNVA (MD 95% CI = 0.20 (0.20; 0.25); *p* < 0.001) and IOP (MD 95% CI = 2.90 (1.55; 2.50); *p* < 0.001) was higher than after 6 months. The level of contrast sensivity was again lower at the baseline than after 6 months (MD 95% CI = −0.10 (−0.20; −0.15); *p* < 0.001)

### 3.3. Evaluation of Dysphotopsia

Halo and glare after 6 months were experienced more often among subjects form the MULTI group than among subjects from the two other groups (65% of eyes in MULTI group vs. 6% of eyes in the EDOF group and 0% of eyes in the MONO group; *p* < 0.001 for halo and 10% of eyes in the MULTI group vs. 3% of eyes in the EDOF group and 0% of eyes in the MONO group; *p* = 0.045). No other significant difference was detected between groups and between the occurrence of photic phenomena 6 months after the procedure, Table 4.

### 3.4. Spectacle Dependence and Patient Satisfaction

Glasses were needed by 35% of subjects from the EDOF group, by 96% of subjects from the MONO group and by no one from the MULTI group (this was a statistically significant dependency—*p* < 0.001). Every patient from every group was satisfied (both for the right and left the eye). Most of subjects from each group rated their satisfaction (both for right and left eye) with a number 5 (97% for right eye and 86% for left eye in the EDOF group; 92% for right eye and 81% for left eye in the MONO group; 77% for right eye and 85% for left eye in the MULTI group; *p* = 0.552 for the dependency between groups and level of satisfaction for right eye and *p* > 0.999 for the dependency between groups and level of satisfaction for left eye). Among the MULTI group, there was a greater proportion of subjects that rated their satisfaction for the right eye with the number 4 than in two other groups (23% vs. 3% in EDOF and 8% in MONO; *p* = 0.038), Table 5.

## 4. Discussion

### 4.1. Qualification

Due to the non-diffractive structure of the lens and one elongated focus, the EDOF lens can be implanted in patients who do not qualify for multifocal lens implantation or are afraid of either the photic phenomena or reduced contrast sensitivity. In the case of multifocal lenses, the eligibility criteria are the strictest, and patients should be free of ocular diseases in order to achieve the best possible vision after surgery. As can be seen in Table 1, the patients who were qualified for cataract surgery with a non-diffractive EDOF lens had eyes with various ocular pathologies or past refractive surgeries, which did not impair the prognosis for improved vision after surgery, and thus patient satisfaction after surgery was high. The profile of patients in this case is similar to that of patients qualified for surgery with monofocal lens implantation.

### 4.2. Postoperative Results

The non-diffractive EDOF lens provides good acuity of distance vision, intermediate distance and functional near vision, confirmed by previous studies [19]. In our study, patients who had the Vivity lens implanted achieved a significant improvement in VA at all distances. The UCDVA at 4 m monocular is similar between the EDOF, the MONO group and the MULTI group. In the case of the monocular UCIVA at 80 cm, VA is better for patients in the EDOF group than for the MONO and MULTI groups (UCIVA EDOF = 0.0, MULTI = 0.1, MONO = 0.3, respectively). In the case of monocular UCNVA, patients in the EDOF group achieved a worse VA than patients in the MULTI group and better than in the MONO group, as confirmed by other published studies comparing EDOF lenses with multifocal and monofocal lenses [15,19,20,21]. The worse monocular UCIVA at 80 cm for the MULTI group compared to the EDOF group may be due to the fact that the PanOptix lens has a focus to intermediate distance at 60 cm [15].

To the best of our knowledge, no study has been published, comparing at the same time the Vivity lens with monofocal and multifocal to date. In a large randomized study, Bal C. et al. compared postoperative outcomes in patients implanted with a Vivity lens compared to an aspheric monofocal lens. They described better intermediate and near distance vision after implantation of the Vivity lens and a similar visual impairment profile compared to the aspheric monofocal IOL [20].

In the study by Kohnen T. et al. the postoperative outcomes after bilateral Vivity lens implantation with target refraction set to emmetropia were assessed (32 eyes—16 patients). Patients achieved: binocular uncorrected VA for distance, intermediate distance and near distance, respectively, 0.01 ± 0.05 logMAR at 4 m, 0.05 ± 0.05 logMAR at 80 cm, 0.07 ± 0.06 logMAR at 66 cm and 0.25 ± 0.11 logMAR at 40 cm [22]. The results obtained in our study, for EDOF lens implantation with target refraction set to emmetropia were very similar (Me [Q1; Q3]): binocular UDCVA at 4 m 0.00 (−0.09; 0.00) logMAR, binocular UCIVA at 80 cm 0.00 (0.00; 0.03) logMAR, and binocular UCNVA at 40 cm 0.30 (0.22; 0.36) logMAR, respectively.

Arrigo A. et al. describes the authors’ own experiences in healthy eyes (108 eyes—54 patients) after EDOF Vivity lens implantation. Very good results of distance vision and intermediate distance were described; in the case of near vision, the need for an addition of at least +1.0 D was indicated. Monocular UCDVA was 0.1 ± 0.04 logMAR, monocular BCDVA was 0.0 ± 0.03 logMAR, respectively [23]. Patients in our study with an implanted Vivity lens also needed a near vision supplement of about 1D or more and median monocular UCDVA was 0.0 logMAR, median monocular BCDVA was 0.0 logMAR.

The use of the minimonovision system in the case of the Vivity lens improves the VA for near vision and increases the degree of independence from ocular correction. In the paper by Newsom T. et al. describing the results of binocular Vivity lens implantation with target of slight myopia −0.75 D, 29 of 33 eyes achieved UCNVA binocular 0.2 logMAR or better [24]. Very similar results were obtained in our patients wih minimovision, whose median binocular UCNVA was 0.2 logMAR.

Rementería-Capelo LA et al. describes the postoperative results after binocular Vivity lens implantation in patients with ocular pathology. A monocular UCDVA was achieved in the test group of 0.03 ± 0.8 logMAR, compared to the control group with an implanted Vivity lens without eye pathology −0.1 ± 0.07. The statistical difference between binocular UCDVA in both groups was not described, as was the case of defocus curves and contrast sensitivity [25]. The result of this study is similar to ours, which evaluated the visual acuity of both healthy patients and those with ocular pathology (median of monocular and binocular UCDVA was 0.0 logMAR). These results, although described on small groups and with a wide range of ocular disorders, give evidence that ocular disorders do not disqualify from Vivity lens implantation. Postoperative results in these patients are very good.

### 4.3. Spectacle Dependance

Therefore, EDOF lenses can be positioned between monofocal and multifocal lenses, they provide good uncorrected visual acuity for distance and intermediate, but uncorrected visual acuity for nearsightedness may be insufficient. In our study, glasses were needed by 35% of subjects from the EDOF group, by 96% of subjects from the MONO group and by no one from the MULTI group (this was a statistically significant dependency—*p* < 0.001). In a study by Rementería-Capelo LA et al., 40% of patients in both study groups (with and without ocular pathology) reported never using close-up glasses [25]. Similar results were described by Kohnen et al. (38%) [22]. The higher degree of independence from spectacle correction among our patients with EDOF lens implantation may be due to the different profile of qualified patients for the procedure. In addition, our EDOF group was not a homogeneous group and some patients had the target set to emmetropia and some had the minimovision system applied. This was due to patient preference and their desire to improve their near vision. In addition, each patient prefers a different reading distance, which also contributes to the different results.

As you know, the use of a minimovision system in the cases of the Vivity lens improves near vision acuity and increases the degree of independence from spectacle correction. Newsom T. et described a high level of satisfaction and a greater degree of independence from ocular correction with the monovision system than without monovision with implantation of the same lens [24].

### 4.4. Occurance of Photic Phenomena

In our study, a small number of patients after EDOF implantation reported photic phenomena (14% patients). Patients who reported the occurrence of dysphotopsia described these side effects as minor, not disrupting normal functioning, similar to the previous reports [20,22,23]. Compared to patients in the MONO group, the incidence of dysphotopsia is slightly more frequent, but it is definitely seen less than in patients in the MULTI group. The study by Rementería-Capelo LA et al. described that patients reported a higher prevalence of halos and glare than other reports on Vivity IOL, especially in the study group, with ocular pathologies: 60% halo, 54% glare, compared to the control group, where the incidence of halo was 28% and 48% [25]. Kohnen et al. found that 25% reported halo and 25% glare [22]. Arrigo et al. reported that 30% and 33% of patients reported halo and glare [23]. The differences between the studies may be due to differences in the questionnaires used in the study and the “inquiry,” an active question about the presence of dysphotopsia, and this has been shown to increase reporting rates [26].

A paper by Newsom T. et described that the use of a monovision system when implanting a Vivity lens compared to target refraction set to emmetropia does not increase the frequency of photic phenomena [24].

### 4.5. Contrast Sensivity

Although this was not the main aim in the study also described was the contrast sensivity. Some patients, due to the fear of decreased of contrast sensivity after surgery, choose not to implant multifocal lenses and select EDOF lenses. According to the manufacturer, the Vivity lens has a safety profile similar to that of monofocal lenses [11,19]. In our study, contrast sensitivity in patients implanted with the Vivity lens did not differ significantly from patients implanted with a monofocal lens and was better than that of patients implanted with a multifocal lens. It is difficult to relate the results of this study to others, since contrast sensitivity was tested only with nearsighted charts and only under photopic conditions.

In our study, we encountered a few limitations. First, it was conducted at a single center, so the number of patients in the study was limited. Secondly, each study group included both healthy patients and patients with eye pathology and after refractive surgery. Studies focusing on specific ocular pathologies or on patients after specific refractive procedures, with larger numbers of patients, would be necessary to best determine which type of IOL would provide the greatest benefit for a given group of patients. In addition, comparing data from our study with other published studies is problematic due to different inclusion/exclusion criteria, study conditions and procedures.

## 5. Conclusions

The majority of patients presenting for cataract surgery who whished to increase independence from spectacle correction are eligible for the implantation of a non-diffractive EDOF lens. Postoperative visual acuity improves at any distance. In the case of the monocular uncorrected intermediate visual acuity at 80 cm, it is better for patients with EDOF lens than with monocofal or multifocal lens. In the case of monocular uncorrected near visual acuity at 40 cm, patients with EDOF lens achieved worse visual acuity than patients with multifocal lens and better than with monofocal lens. The EDOF lens definitely increases independence from spectacle correction compared to monofocal lenses (65% vs 4%); however, the greatest degree of independence from spectacles is provided by multifocal lenses (100%). Only 14% patients after EDOF implantation reported photic phenomena. Compared to patients with monofocal lens, the incidence of dysphotopsia is slightly more frequent, but definitely it is seen less than in patients with multifocal lens.

## Figures and Tables

**Table 1 jcm-11-05246-t001:** Demographics and preoperative characteristics (refractive and monocular VA (logMAR) data, ocular pathologies) of the three groups.

Variables	EDOF n = 70 Eyes n = 35 Subjects	MONO n = 52 Eyes n = 26 Subjects	MULTI n = 52 n = 26 Subjects	*p*	p^1^	p^2^	p^3^
	Me (Q1; Q3) or M ± SD/n (%)						
Sex (female)	23 (65.7)	15 (57.7)	19 (73.1)	0.506	-	-	-
Age	57.74 ± 9.37	60.69 ± 10.65	59.46 ± 8.41	0.439 ^2^	-	-	-
Refractive error							
Myopia	31 (44.3)	27 (51.9)	12 (23.1)	**0.012 ^1^**	-	-	-
Emmetropia	5 (7.1)	2 (3.8)	9 (17.3)				
Hyperopia	34 (48.6)	23 (44.2)	31 (59.6)				
Axial lenght (mm)							
Short < 22	8 (11.4)	4 (7.7)	1 (1.9)	0.070 ^1^	-	-	-
Medium 22–26	57 (81.4)	48 (92.3)	49 (94.2)				
Long > 26	5 (7.1)	0 (0.0)	2 (3.8)				
Toric IOL	19 (27.1)	12 (23.1)	10 (19.2)	0.606	-	-	-
MRSE (D)	0.50 (−2.50; 1.75)	−1.00 (−2.50; 2.31)	1.12 (−0.25; 2.00)	0.329	-	-	-
UCDVA	0.70 (0.30; 1.00)	0.70 (0.40; 0.77)	0.40 (0.29; 0.70)	**0.017**	0.624	**0.015**	**0.010**
BCDVA	0.25 (0.20; 0.30)	0.25 (0.25; 0.32)	0.25 (0.25; 0.25)	**0.004**	**0.005**	0.735	**0.002**
UCIVA	0.75 (0.43; 0.93)	0.60 (0.40; 0.90)	0.70 (0.48; 0.94)	0.236	-	-	-
BCIVA	0.20 (0.10; 0.40)	0.30 (0.10; 0.40)	0.20 (0.10; 0.30)	0.229	-	-	-
UCNVA	0.80 (0.70; 1.15)	0.70 (0.50; 0.86)	0.80 (0.60; 1.02)	0.054	-	-	-
BCNVA	0.30 (0.10; 0.50)	0.40 (0.20; 0.40)	0.20 (0.18; 0.32)	**0.014**	0.131	0.297	**0.001**
Contrast sensivity	1.80 (1.70; 1.90)	1.70 (1.58; 1.80)	1.80 (1.70; 1.83)	0.051	-	-	-
IOP (mmHg)	17.30 (16.25; 17.30)	17.30 (14.60; 17.30)	17.30 (14.60; 17.30)	0.270	-	-	-
ACD (mm)	3.15 (2.94; 3.52)	3.03 (2.80; 3.23)	3.17 (2.89; 3.36)	**0.012**	**0.003**	0.193	0.113
IOL power (D)	22.25 (19.50; 24.00)	23.00 (21.12; 23.62)	23.00 (21.00; 24.50)	0.553	-	-	-
Ocular pathology							
Glaucoma	4 (11.4)	8 (30.8)	2 (7.7)	0.0681	-	-	-
AMD	4 (11.4)	4 (15.4)	1 (3.8)	0.3751	-	-	-
Retinopathy	4 (11.4)	4 (15.4)	0 (0.0)	0.1171	-	-	-
Refractive surgery	4 (11.4)	0 (0.0)	0 (0.0)	0.0371	-	-	-
PEX	2 (5.7)	2 (7.7)	4 (15.4)	0.5261	-	-	-
Drug-induced cataract	8 (22.9)	4 (15.4)	0 (0.0)	0.0251	-	-	-

Qualitative variables were described as n (%) and quantitative variables—as median with quartile 1 and 3 or mean with standard deviations. Dependencies between groups and qualitative variables were made using chi-square test or Fisher’s exact test ^1^. Comparisons of quantitative variables’ level were made with Kruskal-Wallis test or with ANOVA ^2^ analysis. *p*—*p* value for main analyses; *p* value for post-hoc analyses: p^1^—EDOF vs. MONO p^2^—EDOF vs. MULTI, p^3^—MONO vs. MULTI. Abbreviations: mm—milimiters, IOL—intraocular lens, AMD—age-related macular degeneration, PEX—pseudoexfoliation syndrome, MRSE—mean refraction spherical equivalent, D-diopters, mmHg—millimetres of mercury, UCDVA—uncorrected distance visual acuity at 4 m, BCDVA—best corrected distance visual acuity, UCIVA—uncorrected intermediate visual acuity at 80 cm, BCIVA—best corrected intermediate visual acuity, UCNVA—uncorrected near visual acuity at 40 cm, BCNVA—best corrected near visual acuity.

**Table 2 jcm-11-05246-t002:** Comparison of postoperative data (6 months after cataract surgery) between three groups.

Variables	EDOF	MONO	MULTI	*p*	p^1^	p^2^	p^3^
	Me (Min–Max)						
MRSE (D)	−0.25 (−1.25; 0.50)	0.00 (−1.00; 0.75)	0.25 (−0.50; 0.75)	**<0.001**	0.053	**<0.001**	0.221
Monocular visual acuity (logMAR)							
UCDVA	0.00 (−0.20; 0.20)	0.00 (−0.10; 0.40)	0.00 (−0.15; 0.20)	0.433	-	-	-
BCDVA	0.00 (−0.20; 0.00)	0.00 (−0.20; 0.00)	0.00 (−0.15; 0.00)	**0.008**	0.335	**0.009**	0.233
UCIVA	0.00 (−0.10; 0.30)	0.30 (−0.10; 0.60)	0.10 (−0.20; 0.30)	**<0.001**	**<0.001**	**0.004**	**<0.001**
BCIVA	0.00 (−0.10; 0.10)	0.00 (−0.20; 0.20)	0.00 (−0.20; 0.20)	**0.008**	0.590	**0.004**	0.331
UCNVA	0.40 (0.00; 0.60)	0.50 (0.10; 0.90)	0.10 (0.00; 0.20)	**<0.001**	**<0.001**	**<0.001**	**<0.001**
BCNVA	0.00 (−0.10; 0.20)	0.00 (−0.10; 0.10)	0.00 (0.00; 0.30)	**0.008**	0.301	0.229	**0.007**
Binocular visual acuity (logMAR)							
UCDVA	0.00 (−0.15; 0.10)	0.00 (−0.10; 0.14)	0.00 (−0.15; 0.10)	0.767	-	**-**	-
UCIVA	0.00 (−0.10; 0.10)	0.20 (−0.20; 0.40)	0.00 (−0.10; 0.14)	**<0.001**	**<0.001**	**0.015**	**<0.001**
UCNVA	0.26 (0.00; 0.46)	0.40 (0.04; 0.70)	0.00 (0.00; 0.10)	**<0.001**	**<0.001**	**<0.001**	**<0.001**
Contrast sensitivity	2.00 (1.80; 2.00)	2.00 (1.80; 2.00)	1.90 (1.80; 2.00)	**<0.001**	0.907	**<0.001**	0.063

Variables were described as median with range of scores (min–max). Comparisons of quantitative variables’ level were made with Kruskal-Wallis test. *p*—*p* value for main analyses; *p* value for post-hoc analyses: p^1^—EDOF vs. MONO, p^2^—EDOF vs. MULTI, p^3^—MONO vs. MULTI.

**Table 3 jcm-11-05246-t003:** Comparison of uncorrected binocular acuity at all distances between four groups (EDOF group divided into 2 subgroups: emmetropia i minimonovision).

Variables	Emmetropia	Minimonovision	MONO	MULTI	*p* for Main Analyses
	Me (Q1; Q3)				
UCDVA	0.00 (−0.09; 0.00)	0.00 (−0.10; 0.00)	0.00 (−0.06; 0.00)	0.00 (−0.09; 0.00)	0.904
UCIVA	0.00 (0.00; 0.03)	0.00 (−0.06; 0.00)	0.20 (0.10; 0.24)	0.00 (0.00; 0.06)	**<0.001**
UCNVA	0.30 (0.22; 0.36)	0.20 (0.10; 0.30)	0.40 (0.30; 0.50)	0.00 (0.00; 0.03)	**<0.001**
***p* value for post-hoc analyses**					
				UCIVA	UCNVA
Emmetropia vs. minimonovision				**0.040**	0.062
Emmetropia vs. MONO				**<0.001**	**0.001**
Emmetropia vs. MULTI				0.172	**<0.001**
Minimonovision vs. MONO				**<0.001**	**<0.001**
Minimonovision vs. MULTI				**<0.001**	**<0.001**
MONO vs. MULTI				**<0.001**	**<0.001**

Variables were described as median with quartile 1 and 3. Comparisons of quantitative variables’ level were made with Kruskal-Wallis test.

**Table 4 jcm-11-05246-t004:** Comparison of occurrence of photic phenomena 6 months after the procedure between three groups.

Variables	EDOF n = 70 Eyes	MONO n = 52 Eyes	MULTI n = 52 Eyes	*p*
	n (%)			
Halo	4 (5.7)	0 (0.0)	34 (65.4)	**<0.001 ^2^**
Halo level				
1	1 (25.0)	-	13 (38.2)	0.563
2	3 (75.0)	-	12 (35.3)	
3	0 (0.0)	-	8 (23.5)	
4	0 (0.0)	-	1 (2.9)	
5	0 (0.0)	-	0 (0.0)	
Glare	2 (2.9)	0 (0.0)	5 (9.6)	**0.045**
Glare level				
1	0 (0.0)	-	4 (80.0)	0.143
2	1 (50.0)	-	1 (20.0)	
3	1 (50.0)	-	0 (0.0)	
4	0 (0.0)	-	0 (0.0)	
5	0 (0.0)	-	0 (0.0)	
Starburst	4 (5.7)	2 (3.8)	8 (15.4)	0.097
Starburst level				
1	1 (25.0)	0 (0.0)	2 (25.0)	0.899
2	3 (75.0)	2 (100.0)	4 (50.0)	
3	0 (0.0)	0 (0.0)	2 (25.0)	
4	0 (0.0)	0 (0.0)	0 (0.0)	
5	0 (0.0)	0 (0.0)	0 (0.0)	

Variables were described as n (%). Dependencies between groups and qualitative variables were made using chi-square test ^2^ or Fisher’s exact test.

**Table 5 jcm-11-05246-t005:** Number of patients needing glasses and satisfaction rating broken down by groups.

Variable	EDOF n = 35 Subjects	MONO n = 26 Subjects	MULTI n = 26 Subjects	*p*
	n (%)			
Glasses	14 (35.0)	25 (96.2)	0 (0.0)	**<0.001 ^2^**
Satisfaction (right eye)	35 (100.0)	26 (100.0)	26 (100.0)	-
Satisfaction (left eye)	35 (100.0)	26 (100.0)	26 (100.0)	-
Level of satisfaction (right eye)				
1	0 (0.0)	0 (0.0)	0 (0.0)	**0.038**
2	0 (0.0)	0 (0.0)	0 (0.0)	
3	0 (0.0)	0 (0.0)	0 (0.0)	
4	1 (2.9)	2 (7.7)	6 (23.1)	
5	34 (97.1)	24 (92.3)	20 (76.9)	
Level of satisfaction (left eye)				
1	0 (0.0)	0 (0.0)	0 (0.0)	0.932
2	0 (0.0)	0 (0.0)	0 (0.0)	
3	0 (0.0)	0 (0.0)	0 (0.0)	
4	5 (14.3)	5 (19.2)	4 (15.4)	
5	30 (85.7)	21 (80.8)	22 (84.6)	

Dependencies were calculated using chi-square test ^2^ or Fisher’s exact test.

## Data Availability

Not applicable.

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
