# Peer review of "Evaluation of Visual and Patient—Reported Outcomes, Spectacle Dependence after Bilateral Implantation with a Non-Diffractive Extended Depth of Focus Intraocular Lens Compared to Other Intraocular Lenses"

_jcm, 2022, doi:10.3390/jcm11175246_

Round 1

Reviewer 1 Report

In this work the authors evaluated clinical results of cataract surgery in patients implanted with the Vivity IOL in comparison with other multifocal and monofocal IOLs.

A list of issues to be corrected is as follows:

-         1- L14 (and L 84). Use the key words “photic phenomena”.  Include a reference In L 84.

-         2- L 19: Describe better the control groups: “52 eyes in 26 patients in whom a multifocal lens or a monofocal lens was implanted” is not clear enough.

-         3-  L 74: Please include a Reference for IC-8  and IOL

-         4- L 146 a 153: Please include a Reference (another paper and/or data sheet) for Panoptix IOL.

-         5- L 206: Statistics: Please justify the use the S-W test.

-         6- L 233: erratum: “For myopia there was a spherical equivalent less than 0.5 D”

-         7- Please check the negative value of CS (last line) in Table 8.

-         8- L 387: eyed à eyes

-         9- Consider to redo Table 10. It seems unnecessarily large. Just giving the mean values, SD and p-values  would be enough.

-         10- L 428: Ref [17] or [16]?, check the order.

-         11- L 433: 80 cm, VA?

-         12 L- 437: Please include references.

-         13 L-  438: indirect?

-         14 L- 456 to 464: Please compare the results of Ref. [20] with the results of the present study.

-         15- Please discuss the obtained CS values.

-         16- References [11], [12]. Please include full info:  Title of Site. Available online: URL (accessed on Day Month Year.

-         17- Rewrite the Conclusions avoiding the terms “most” , majority” “a small number”. Instead, use percentages.

Author Response

Thank you very much for your review. I have responded to the points mentioned.

-         1- L14 (and L 84). Use the key words “photic phenomena”.  Include a reference In L 84.

Thank you, I changed it.

-         2- L 19: Describe better the control groups: “52 eyes in 26 patients in whom a multifocal lens or a monofocal lens was implanted” is not clear enough.

 Thank you for your attention, I added more information.

-         3-  L 74: Please include a Reference for IC-8  and IOL

I added. Thank you.

-         4- L 146 a 153: Please include a Reference (another paper and/or data sheet) for Panoptix IOL.

Thank you, I changed.

-         5- L 206: Statistics: Please justify the use the S-W test.

The analysis of whether the distribution is normal is needed in order to choose the appropriate methods for further analysis.  As shown in numerous simulation studies, the Shapiro-Wilk test is the most powerful test regardless of the sample size (Razali & Wah, 2011, Wijekularantha et al., 2019).

Razali, N. M., & Wah, Y. B. (2011). Power comparisons of shapiro-wilk, kolmogorov-smirnov, lilliefors and anderson-darling tests. Journal of statistical modeling and analytics, 2(1), 21-33.

Wijekularathna, D. K., Manage, A. B. W., & Scariano, S. M. (2019). Power analysis of several normality tests: A Monte Carlo simulation study. Communications in Statistics - Simulation and Computation, 1–17. doi:10.1080/03610918.2019.1658780

-         6- L 233: erratum: “For myopia there was a spherical equivalent less than 0.5 D”

Thank you, I corrected it.

-         7- Please check the negative value of CS (last line) in Table 8.

Thank you, I corrected it. It was my mistake.

-         8- L 387: eyed  eyes

Thank you, I corrected it.

-         9- Consider to redo Table 10. It seems unnecessarily large. Just giving the mean values, SD and p-values  would be enough.

As for Table 10, I purposely did not give SD and p-values, because some groups are simply too few to count such statistics.

-         10- L 428: Ref [17] or [16]?, check the order.

Thank you for your attention, I have corrected the order.

-         11- L 433: 80 cm, VA?

I have corrected this sentence.

-         12 L- 437: Please include references.

Thank you, I corrected it.

-         13 L-  438: indirect?

That's my mistake, it was about intermediate.

-         14 L- 456 to 464: Please compare the results of Ref. [20] with the results of the present study.

Thank you, I did.

-         15- Please discuss the obtained CS values.

Thank you, I did.

-         16- References [11], [12]. Please include full info:  Title of Site. Available online: URL (accessed on Day Month Year.

Thank you, I did.

-         17- Rewrite the Conclusions avoiding the terms “most” , majority” “a small number”. Instead, use percentages.

Only in the case of the majority term, referring to those eligible for cataract surgery with implantation of the EDOF non-diffractive lens, a numerical value was not given, as it is difficult to determine in percentage terms which patients qualify for implantation of this lens. These are always individual decisions of the doctor and the patient.

Thank you again for the review and the very concrete information on things that should be improved or discussed more extensively.

Kind regards,

Anna Dołowiec-Kwapisz

Reviewer 2 Report

This is a paper that aims to evaluate the different clinical and patient-reported outcomes comparing the use of extended depth 15 of focus (EDOF) intraocular lens compared with mono-focal and multifocal lenses.

While the authors provided a comprehensive introduction to their investigation, there were major flaws in reporting the methodology and results. In short, the study lacked cohesion between the aims as stated by the authors and the way the results were portrayed. In addition, there were major redundancies in the analysis of the findings and the numerous tables used which could be extensively improved and summarized. It would be best for the authors to focus on the main comparisons being investigated so that it does not get diluted with various and redundant sub-analyses.

Kindly find the suggestions for the major issues that need to be addressed:

-          Section 2.4 is out of place in the methods section and dilutes the sequence in the methodology as the reader is following it. It would be best to summarize the descriptions of the different lenses into a table (ideally supplementary) so that it is not in the way of the natural flow of a methods section. In addition, more references need to be added to certain descriptions that the authors mention to the different lenses.

-          Results section requires major alterations that focus specifically on the outcomes that are in the aims of this investigation. The authors add a variety of sub-analyses, many of which are redundant, that dilute the main message or “punch-line” of the study and confuse the reader as they follow the flow of the manuscript. For example, there is no need to perform a separate comparison between baseline vs 6 months for each lens when, immediately after, the authors then compare different postoperative time points, including the 6 months again, and later the 6-month outcomes are re-listed when the 3 groups are being compared. This is disruptive, especially considering that the main aim of the study is the comparison of the 3 groups and not the comparison to the baseline. The individual analysis of the 3 lenses should be summarized in the text or grouped into 1 table altogether so as to not dilute the results of the 3 group comparison that follows.

-          Overall the number of tables is overwhelming and needs to be reduced, especially considering the redundancy in many of them. These results could either be removed or summarized in the text as a secondary sub-analysis.

-          The discussion lacks a comprehensive interpretation of the results with regard to the findings in the literature and requires extensive changes. The authors also fail to fully address the limitations of the study in order for the reader to keep them in mind when drawing conclusions.

-          The conclusion section requires improvement in the flow of how the major outcomes are summarized.

Other suggestions:

-          Introduction: add a section about what is missing in the literature that led to the concept of this study. And how addressing these limitations would add to the literature and improve patient care.

-          Section 2.2: EDOF group sample size is listed whereas the other 2 are not. Either list the sample size of all 3 groups or none as they are listed in the results eventually.

-          Lines 230-236 should not be in the results as they are part of the methodology of the study.

Author Response

Thank you very much for your reviews. I have tried to address all the listed information that should be changed or better discussed in this paper.

Responding to the listed points one by one, I have changed:

- Section 2.2 - the size of all groups is mentioned (EDOF, MONO, MULTI).

- Section 2.4 - I dropped the description of the lenses, putting one sentence about them next to the information about what lenses were implanted in the patients in the study. The Vivity lens was discussed more extensively in the introduction, where information was given on why this lens stands out from other EDOF lenses.

- Results - in line with your suggestions, I focused on the main objectives of the study, and information on the exact analyses for specific lenses was removed. I added only one subanalysis, without a table, summarizing how baseline parameters improved 6 months after treatment relative to baseline in each group.  This significantly reduced the number of tables and descriptions.

- The summary section was described in more detail to address the main objectives of the study.

- Line 230-236 was moved to methodology as suggested.

- The introduction referred to why the article was written and what was missing from the literature.

- The discussion has been extensively revised. Subsections have been added with answers to all the main objectives of the study, in order to organize the discussion. I also wrote about the limitations of the study and what could be investigated in the future.

Thank you again for the review.

Kind regards,

Anna Dołowiec-Kwapisz